# Effect of Quinolones Versus Cefixime on International Normalized Ratio Levels After Valve Replacement Surgery with Warfarin Therapy

**DOI:** 10.3390/medicina55100644

**Published:** 2019-09-26

**Authors:** Anam Liaqat, Arif-ullah Khan, Muhammad Asad, Hafsa Khalil

**Affiliations:** 1Riphah Institute of Pharmaceutical Sciences, Riphah International University, Islamabad 44000, Pakistan; anamliaqat29@gmail.com; 2Armed Forces Institute of Cardiology and National Institute of Heart Diseases, Rawalpindi 46000, Pakistan; masad1288@gmail.com (M.A.); hafsakhalil100@gmail.com (H.K.)

**Keywords:** warfarin, drug interaction, moxifloxacin, levofloxacin, cefixime

## Abstract

*Background and Objectives:* A dispute over interaction of warfarin with two quinolones—i.e., moxifloxacin and levofloxacin—leading to significant increase in international normalized ratio (INR) levels and coagulopathies is currently in debate. The study objective was to compare the INR values due to addition of quinolones and cefixime in warfarin treated patients after replacement of disease valves with metallic valves. *Material and Methods:* A prospective evaluation of patients who undergone valve replacement surgeries in the cardiology hospital setup in Pakistan during the period 2018–2019 was done, including all those subjects treated concurrently with levofloxacin, moxifloxacin, cefixime, and warfarin for the study. Data organized included demographic information, concurrent medications, and appropriate analytical parameters, especially INR values taken before and within seven days after prescribing three antibiotics in discharged patients who had undergone valve replacement surgeries. Patients for whom laboratory INR values were not given at the time of discharge and with deranged liver function, renal function, low albumin levels, and febrile patients were removed from study. Furthermore, patients were advised on possible food interactions and evaluated to examine if these factors have any possible influence on the interaction being studied. *Results:* Differences in INR were analyzed statistically by means of SPSS analysis before and after the possible interaction. Following the administration of levofloxacin and moxifloxacin to warfarin therapy, statistical analysis showed remarkable increase in INR (*p* < 0.001) and no significant change in INR was observed after cefixime treatment (*p* > 0.05). *Conclusion:* Results showed that, after adding levofloxacin and moxifloxacin in patients on warfarin, therapy contributed to remarkable increase in INR. However, addition of cefixime prevented frequent coagulopathies; therefore, close monitoring of INR and switching to a safe antibiotic such as cefixime is recommended.

## 1. Introduction

The arrival of valve replacement surgery in the early 1960s has dramatically enhanced the outcomes of valvular heart disease patients. About 90,000 valve replacements are now implanted in the United States and 280,000 are implanted globally each year, about half are mechanical valves and half are bioprosthetic valves. Prosthetic valve patients are at danger of thromboembolic complications, including systemic embolization, most frequently cerebral and prosthetic thrombosis, ultimately causing blockage and/or regurgitation of the valve. The danger of thromboembolic occurrences is greater for mechanical than for bioprosthetic valves, greater for mitral than for aortic prosthetic valves, and greater for early (<3 months) versus late postoperative stage. In the presence of concomitant thromboembolism risk variables, including atrial fibrillation, left ventricular dysfunction, left atrial dilation, prior thromboembolism, and hypercoagulable disease, the risk is also increased. Due to the risk of thromboembolism, mechanical prosthetic valve patients require long-term anticoagulant therapy, which should be initiated as soon as possible after valve replacement, preferably within 6 to 12 h. Vitamin K antagonists (VKA), usually for long-term treatment and heparin (primarily non-fractionated or low molecular weight heparin for short term bridge therapy), are the anticoagulants used to avoid valve thrombosis and thromboembolic occurrences in patients with prosthetic heart valves [1].

In 1954, Warfarin was licensed for human use in the United States and it is indicated for both primary and secondary prevention of thromboembolic disease. Various factors such as small therapeutic range and its interactions with various drugs and food items lead to limitations in its utilization [2]. By interfering with vitamin K redox cycle, warfarin reduces the production of functional vitamin K depended clotting factors such as factors II, VII, IX, X, protein C, and protein S [3]. It is indicated in conditions like deep venous thrombosis, pulmonary embolism, prophylaxis, and treatment of systemic embolic complications such as stroke associated with atrial fibrillation, post myocardial infarction, rheumatic valve disease, antiphospholipid syndrome, and after cardiac valve replacement surgeries. The Federal Drug Administration’s warfarin use indications include long-term anticoagulation after a thrombotic event or prevention of thrombotic event in high-risk patients, including post-operative states, atrial fibrillation, and those with artificial valves.

The traditional method of anticoagulation is use of warfarin, which requires frequent blood tests to monitor prothrombin time (PT) and international normalized ratio (INR) levels. Those with supra-therapeutic INR are at higher risk for presentation with hemorrhagic complications, which includes life-threatening gastrointestinal, intracerebral, and other major bleeding [4,5]. Bleeding in post-surgical patients can be categorized to insignificant, mild, moderate, severe, and massive according to universal definition of perioperative bleeding as shown in Table 1 [6]. The American College of Cardiology and the American Heart Association (ACC/AHA) provides guidelines indicating the appropriate level of anticoagulation in patients after valve replacement surgery depending on the type of valve: Mechanical or biological. The ACC/AHA guidelines on anti-coagulation for prosthesis recommends warfarin use to achieve an INR of 2.0 to 3.0 after replacement of aortic valves with mechanical prostheses. After replacement of the mitral valve by a mechanical valve, warfarin is indicated to achieve an INR goal from 2.5 to 3.5 [7].

Warfarin is known to interact with many drugs, leading to unwanted results. In particular, hemorrhage is a major concern when warfarin is used in conjunction with the other drugs that interact. Metabolism of warfarin occurs via cytochrome P450 enzyme system and several drugs that undergo cytochrome P450 metabolism and therefore, interact with warfarin, ultimately leading to an increase in INR levels. All prosthetic valve patients need adequate antibiotic prophylaxis to prevent endocarditis [8].

Quinolones is one of the commonly prescribed medicine for infection in post-surgery. Levofloxacin is a drug having a broader spectrum activity for Gram-positive and Gram-negative bacteria. Moreover, levofloxacin shows activity against certain atypical organisms (e.g., *Legionella pneumophila*, *Chlamydia pneumoniae*) and also against certain anaerobes as well. Similarly, moxifloxacin has also a large spectrum against Gram-positive and Gram-negative bacteria. It has a relatively benign adverse effect profile, and allows for once-daily dosing [9]. Although literature does not suggest any relevance of drug interaction of moxifloxacin and levofloxacin with warfarin but there are few case reports and series depicting a major drug interaction [10]. It has been reported that potential mechanisms of interaction of quinolones with warfarin include depletion of vitamin K-producing gut flora and displacement of warfarin from plasma albumin binding sites, leading to increase in free plasma concentration of warfarin and significant increase in INR levels [11].

## 2. Material and Methods

The prospective study was conducted in Pakistan between 2017–2019 in the settings of Armed Forces Institute of Cardiology and National Institute of Heart Diseases (AFIC/NIHD) Rawalpindi, in post-operative surgical wards and surgical OPD. All subjects gave their informed consent for inclusion before they participated in the study. The study was conducted in accordance with the Declaration of Helsinki and the protocol that was approved by the ethical committee of cardiac hospital AFIC/NIHD as per AFIC-IERB-SOP-15 (reference no. 22/11/R&D/2018/04). Appropriate sample sizes were taken in a defined duration of study using non-probability consecutive sampling technique. Post-operative patients of both genders (20–65 years) with mitral valve replacement (MVR), double valve replacement (DVR), and aortic valve replacement (AVR) due to various valvular diseases discharged on warfarin therapy were chosen for study. Patients with serum creatinine levels that are higher than normal i.e., >1.2 mg/dL or with alanine aminotransferase (ALT) levels >41 U/L which may contribute to an increase in INR levels, patients receiving complex regimens including drugs that have major interaction with warfarin [12], as well as febrile patients with suspicion of infection, were excluded from the study. Also they were not permitted to use any other over-the-counter concomitant drugs within 7 days of the first dose of warfarin and throughout the research and were not permitted to consume alcohol, grapefruit, caffeine, licorice, quinine, or cranberry-containing foods or drinks during the research until after the outpatient visit on day 7. Eight patients whose INR values were not available at the time of discharge were removed from the study. The remaining 75 patients were placed in three groups based on antibiotics prescribed after valve replacement surgery. All 75 patients discharged with warfarin received standard 5 mg dosages. Group A patients on warfarin therapy after mechanical valve replacement receiving quinolone such as moxifloxacin while group B patients on warfarin therapy received levofloxacin Group C members on warfarin therapy were not receiving quinolones but receiving another antibiotic, i.e., cefixime. A documented therapeutic INR before the start of quinolones and cefixime at the time of discharge and a documented INR within 7 days after it has been started was recorded and results were compared by means of SPSS analysis. The following data were compiled through clinical workstation consultation and clinical record review, such as demographic information (age, gender), concomitant medications, toxic habits such as smoking and alcohol, as well as the analytical parameters concerned such as creatinine, albumin, ALT, and INR. Particular emphasis placed on these INR before and after possible interaction.

## 3. Results

Total *n* = 75 patients i.e., 46 males and 29 females, (61.3% and 38.7% respectively) with age range of 24–72 and having mean ± standard deviation (SD) of 47 ± 12.2 years after valve replacement surgery completed the study (Table 2). While, moxifloxacin group of patients were having ages between 24–63 y, cefixime group patients between 29–64 y and levofloxacin group between 28–72 y. Total 12 patients with aortic stenosis (AS), 3 with mitral regurgitation (MR), 2 aortic stenosis and aortic regurgitation (AS/AR), 2 aortic stenosis and mitral stenosis(AS/MS), 4 MS, 1 AR, and 1 MS/MR were presented in moxifloxacin group. In cefixime group 11 patients were AS, 1 with AS/MS, 1 with MR, 8 MS, 1 with MS/MR, and 3 AR. Levofloxacin group patients were 11 with AS, 10 MS,1 AS/MR, 1 AS/MR/AR,1 MR, and 1 AS/MS.

Patients with different valvular diseases undergone MVR (presented with MS, MR, and MS/MR), AVR (AS, AR, AS/AR patients) and DVR (AS/MR, AS/MS, and AS/MR/AR patients) surgeries with the metallic valve and therefore they required lifelong anticoagulation therapy with warfarin indication. Most of the subjects under study were having heart valve disease with aortic valve and undergone with AVR, and included 54% patients followed by MVR with 39% patients and finally 7% subjects with DVR having mixed aortic and mitral valve diseases (Table 3).

Frequency and percentages of subjects with various valvular diseases such as patients presented with aortic stenosis, mitral stenosis, mitral valve regurgitation, aortic valve regurgitation, and with mixed valvular disease are given in Table 4.

All patients received standard 5 mg of Warfarin dose before administration of three antibiotics—i.e., moxifloxacin, levofloxacin, and cefixime—while discharged, with INR values in between normal therapeutic range. Total duration of INR monitoring was within 7 days after the administration of three antibiotics in discharged patients. All patients receiving warfarin had normal liver function, albumin levels and serum creatinine levels (with their mean **±**SD values are given in Table 5) to meet the exclusion criteria.

Therefore, these factors did not account for changes in INR values in studied individuals. There was a significant difference between the INR values obtained before and within 7 days of beginning of moxifloxacin and levofloxacin administration using SPSS analysis and revealed remarkable increase in INR levels, i.e., 2.48 ± 0.5 vs. 11.9 ± 3.11, and 2.68 ± 0.43 vs. 10.5 ± 3.1 respectively. *p* value after paired sample *t*-test came out to be <0.001 in both cases and showed that results are statistically highly significant. Patients were reported with coagulopathies after administration of moxifloxacin or levofloxacin and required reversal of supra therapeutic INR values which is usually performed with vitamin K in the absence of bleeding or in case of minor but also major bleeding, while fresh frozen plasma (FFP) is usually indicated in case of supra therapeutic INR values with major bleeding. However, no significant increase in INR values was seen within 7 days after initiation of cefixime antibiotic during the study period with mean ± SD 2.58 ± 0.49 vs. 2.78 ± 0.5 whereas *p* value came out to be > 0.05 after statistical analysis (Table 6).

## 4. Discussion

Chronic valvular heart disease is still common in this country mainly among indigenous people with lower economics status, resulting in thickening of mitral and aortic valve leaflets or aortic and mitral valve stenosis. This can be treated with balloon valvotomy or surgical repair either in the form of a mechanical valve or bio prosthetic valve that contributes to continuous improvement of operational results [13,14]. Morbidity and mortality after valve replacement surgery is commonly caused by thrombotic and embolic complications, as well as due to bleeding associated with anti-coagulation therapy [15,16]. During the first three months of valve replacement surgery, the risk associated with thromboembolic complication is usually highest, with the continuous long lasting life time risk for mechanical valve treated patients. In addition, occurrence of atrial fibrillation in most patients, after the replacement of the metallic mitral valve, requires lifelong anticoagulation. These considerations account for the need to address proper techniques of anticoagulation in order to decrease postoperative thrombotic complications.

Warfarin is an oral anticoagulant and mainly acts by depleting functional vitamin K-dependent protein factors—i.e., II, VII, IX, and X—used in blood clotting processes. These factors are formed precursory in the liver and activated by the carboxylation of specific glutamic acid residues, which also require vitamin K as a cofactor in its reduced form. Warfarin activity is determined by the blood test known as INR. The most recent guidelines published in 2017 provides indication of anticoagulation with a vitamin K antagonist (VKA) to achieve an INR of 2.5 recommended for patients with a mechanical bileaflet or current-generation single-tilting disc AVR with no risk factors for thromboembolism (class I B). Anticoagulation with a VKA is indicated to achieve an INR of 3.0 in patients with a mechanical AVR and additional risk factors for thromboembolic events (AF, previous thromboembolism, LV dysfunction, or hypercoagulable conditions) or an older-generation mechanical AVR (such as ball-in-cage). While, the 2014 recommendation remains current (class I B). Anticoagulation with a VKA is indicated to achieve an INR of 3.0 in patients with a mechanical MVR (class I B) [17].

Warfarin’s therapeutic index is very narrow and its anticoagulation effect is modified by numerous factors, such as drugs and food interactions as well as medical conditions of patients which are on warfarin therapy. The average duration to detect an elevated hypoprothrombinemia effect of warfarin is nearly around 5.5 days [18]. Any factor that alters warfarin’s pharmacodynamics and/or pharmacokinetics can result in either sub therapeutic or supra therapeutic INR levels which ultimately contributes to serious repercussions, such as new thromboembolic or bleeding events [19].

Bleeding is the most significant clinical complication associated with the interactions between warfarin with other drugs. The primary factor influencing the risk of bleeding is anticoagulation intensity, and it has been reported that the risk of major bleeding (i.e., intracranial and intraperitoneal) increases when the INR is above 4. Overall, complications of hemorrhage increase sharply if INR exceeds 5, requiring hospitalization. Because warfarin is metabolized by the hepatic microsomal enzyme system mainly by the CYP2C9 enzyme, it is predisposed to a wide range of interactions with drugs. AHA recommends antibiotic prophylactic regimens in patients with underlying high-risk cardiac conditions associated with a cardiac prosthetic valve. Different antimicrobials—such as azithromycin, erythromycin, cimetidine, fibrates, amiodarone, ciprofloxacin, ofloxacin, and sulfamethoxazole-trimethoprim—are known to show drug–drug interactions with warfarin, leading to higher INR values along with increased risk of complications associated with bleeding [20,21].

Most of the patients included in the study were middle-aged and elderly people who were subjected to mechanical aortic or mitral valve replacement surgery or both valve surgery. For various baseline diseases, they also received pharmacological agents. Captopril, nitroglycerin, famotidine, spironolactone, digoxin, and bisoprolol were the other drugs prescribed. Concomitant treatment might affect the development of an episode of bleeding in these patients, but these drugs did not show to interact with warfarin. Significant increase in INR values was noted when those patients taking warfarin anticoagulant treatment after valvular surgery were initiated on a levofloxacin and moxifloxacin antibiotic regimen and manifested with epistaxis and purpura. However, no major differences in INR values were observed within 7 days of cefixime administration following the discharge of patients on warfarin, and they were having INR values in the therapeutic range during their follow-up in OPD. In order to reverse the effect of warfarin and return the INR values to the desired goal, patients reporting coagulopathies due to moxifloxacin and levofloxacin addition were treated with FFPs.

Many elderly patients undergoing warfarin therapy with other serious medical problems—such as fever, hepatic, or renal diseases, all of which contribute to extreme hypoalbuminemia—pose a higher risk of coagulopathy when being treated with warfarin. All three groups of patients selected therefore had normal levels of serum creatinine, test of liver function, and levels of serum albumin. Patients were also divided into two subgroups to check the potential impact of toxic habits such as (smoking) on the interaction. It is noticed that, patients under study with or without smoking experienced elevation in INR with moxifloxacin and levofloxacin therapy, but not with cefixime therapy. Hence, toxic habits showed no major impact on the clinical presentation of the interaction in this analysis.

Moxifloxacin antibiotic exhibits bactericidal activity towards Gram-positive and Gram-negative aerobes and is commonly prescribed for community-acquired pneumonia, acute exacerbation of chronic bronchitis, acute sinusitis, and soft tissue infection. Fluoroquinolone antibiotics, such as levofloxacin, also exhibit antimicrobial activity to target Gram-positive species such as *Streptococcus pneumoniae* and *Enterococcus faecalis*, as well as against Gram-negative bacteria such as *Haemophilus influenza* and *Pseudomonas aeruginosa* [22]. Levofloxacin also exhibits activity against atypical microorganisms and some anaerobes such as *Clostridium perfingens*.

In order to explain the interaction between warfarin and fluoroquinolones, three potential mechanisms should be considered: first, fluoroquinolones lead to cytochrome P450 inhibition of warfarin metabolism. Among fluoroquinolones, however, moxifloxacin is a new generation of quinolone not metabolized by the cytochrome P450 system and metabolized uniquely by glucuronide and sulfate conjugation. Thus, through this mechanism, theoretically moxifloxacin will not interact with warfarin. However, levofloxacin decreases the metabolism of warfarin by inhibiting major metabolizing enzyme cytochrome P450 2C9. Second, moxifloxacin and levofloxacin inhibit the production of vitamin-K-producing gut flora, leading to the depletion of vitamin-K dependent coagulation factors. Thirdly, by removing warfarin from its plasma protein binding site, moxifloxacin and levofloxacin may cause a transient elevation of free warfarin levels [23].

The onset of interaction between warfarin and fluoroquinolones appear to be delayed. It has been reported that ciprofloxacin and ofloxacin shows interaction with warfarin, leading to elongation in PT. Although few case studies including two new cases of warfarin–moxifloxacin interactions were reported and added to the previously published 12 case reports and therefore highlight the importance of such interaction. No statistically remarkable variations in the INR values observed before and those after levofloxacin administration in one study were found in correlation with other studies [24,25,26]. However, in 2009 the retrospective assessment of potential interaction between levofloxacin and warfarin was conducted and a correlation was found between elevation of INR values because of levofloxacin addition to warfarin therapy but does not clarify severity of drug interaction. Furthermore, this was demonstrated by a small number of patients studied and changes in INR values resulting from various variables—such as type of diet, concurrent drug therapy, abnormal liver function tests, low albumin levels, fever, illness, and stress—that could affect warfarin pharmacokinetics or pharmacodynamics. Due to clinical severity of this interaction it is advisable to switch levofloxacin and moxifloxacin with alternative antibiotic in patients on warfarin, such as cefixime as indicated in our study. Also, more frequent monitoring of INR in these patients may be warranted to avoid possible bleeding episodes. However, the impact of genetic factors on warfarin response cannot be overlooked. Indeed, CP2C19*2 and -*3 and VKORC1 G-1639A polymorphisms are strongly associated to warfarin sensitivity and bleeding risk. Such genetic variants are even more important in presence of drug–drug interactions. Therefore, this study confers limitation in this regard and further studies are needed taking consideration of genetic influences on warfarin response [27,28].

## 5. Conclusions

A correlation between adding levofloxacin and moxifloxacin to warfarin therapy and elevating INR values was found in this study which causes frequent coagulopathies in patients on warfarin therapy. Therefore, use of alternative and safe antibiotic such as cefixime is recommended to avoid bleeding complications and deaths due to frequent coagulopathies.

## Figures and Tables

**Table 1 medicina-55-00644-t001:** Universal definition for perioperative bleeding.

Bleeding Definition	Sternal Closure Delayed	Postoperative Chest Tube Blood Loss within 12 h (mL)	PRBC (units)	FFP (units)	PLT (units)	Cryoprecipitate	PCCs	rFVIIa	Reexploration/Tamponade
Class 0 (insignificant)	No	<600	0 *	0	0	No	No	No	No
Class 1 (mild)	No	601–800	1	0	0	No	No	No	No
Class 2 (moderate)	No	801–1000	2–4	2–4	Yes	Yes	Yes	No	No
Class 3 (severe)	Yes	1001–2000	5–10	5–10	N/A	N/A	N/A	No	Yes
Class 4 (massive)	N/A	>2000	>10	>10	N/A	N/A	N/A	Yes	N/A

PRBC, packed red blood cells; FFP, fresh frozen plasma; PLT, platelet concentrate; PCCs, Prothrombin complex concentrate; rFVIIa, Recombinant activated factor VIIa; N/A, not applicable. * Correction of preoperative anemia or hemodilution only; the number of PRBCs used should only be considered in the universal definition of perioperative bleeding when accompanied by other signs of perioperative bleeding.

**Table 2 medicina-55-00644-t002:** Frequency and percentages of male and female gender in the study.

Gender	Frequency	Percentages
Male	46	61.3%
Female	29	38.7%

**Table 3 medicina-55-00644-t003:** Percentages of patients undergone different valve replacement surgeries.

Procedures	Percentages
Aortic valve replacement	54%
Mitral valve replacement	39%
Double valve replacement	7%

**Table 4 medicina-55-00644-t004:** Frequency and percentages of valvular diseases in the patients under study.

Valvular Disease Type	Frequency
Aortic stenosis	45.3%
Mitral stenosis	29.3%
Mitral regurgitation	6.7%
Aortic regurgitation	5.3%
Aortic stenosis, mitral stenosis	5.3%
Mitral stenosis, mitral regurgitation	2.7%
Aortic stenosis, mitral regurgitation, aortic regurgitation	1.3%
Aortic stenosis, mitral regurgitation	1.3%
Aortic stenosis, aortic regurgitation	1.3%

**Table 5 medicina-55-00644-t005:** Biochemical variables of subjects under study at the time of discharge.

Patient Groups Reference Values	Serum Creatinine (0.5–1.2 mg/dL)	Alanine Transferase (7–55 U/L)	Albumin (35–55 g/L)
moxifloxacin + warfarin	0.7268 ± 0.1577	32.72 ± 10.10	43.28 ± 5.828
levofloxacin + warfarin	0.8644 ± 0.1735	27.52 ± 10.85	44.28 ± 6.755
cefixime + warfarin	0.7560 ± 0.1751	34.64 ± 15.45	44.52 ± 6.801

**Table 6 medicina-55-00644-t006:** International normalized ratio (INR) values before administration of antibiotics, i.e., moxifloxacin, levofloxacin, and cefixime and INR values within 7 days after administration of three antibiotics respectively in patients discharged with warfarin therapy.

INR Without/With Antibiotic Therapy	Mean ± S.D.	
INR before moxifloxacin addition	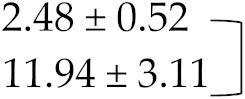	*p* < 0.001
INR within 7 days of moxifloxacin addition			
INR before levofloxacin addition	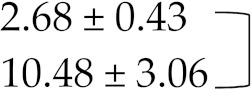	*p* < 0.001
INR within 7 days of levofloxacin addition			
INR before cefixime addition	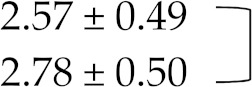	*p* > 0.05
INR within 7 days of cefixime addition			

Paired sample *t*-test was applied between groups and the difference between INR values before and after moxifloxacin and levofloxacin therapy was significant statistically with *p* < 0.001 in both cases. However, no difference between INR values before and after cefixime administration was noted with *p* > 0.05.

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
