# Peer review of "Effect of Quinolones Versus Cefixime on International Normalized Ratio Levels After Valve Replacement Surgery with Warfarin Therapy"

_medicina, 2019, doi:10.3390/medicina55100644_

Round 1
Reviewer 1 Report
This study highlits the importance to monitor factors influencing the response to warfarin, such as drug-drug interactions.
The study is interesting and well written exept for some typos that have to be corrected.
However,
The authors should introduce some information regarding the impact of genetic factors on warfarin response. Indeed, CP2C19*2 and -*3 and VKORC1 G-1639A polymorphisms are strongly associated to warfarin sensitivity and bleeding risk. Such genetic variants are even more important in presence of drug drug interactions.
Therefore, the pharmacogenetic issue should be introduced as a study limitation or at least in the discussion.
The Authors may introduce references in this field:
Mazzaccara et al. Warfarin anticoagulant therapy: a Southern Italy pharmacogenetics-based dosing model. PLoS One. 2013 Aug 26;8(8):e71505. doi: 10.1371/journal.pone.0071505. eCollection 2013.
Johnson et al. Clinical Pharmacogenetics Implementation Consortium Guidelines for CYP2C9 and VKORC1 genotypes and warfarin dosing. Clin Pharmacol Ther. 2011 Oct;90(4):625-9. doi: 10.1038/clpt.2011.185.
Author Response
Thank you for your comments
Point 1: The authors should introduce some information regarding the impact of genetic factors on warfarin response. Indeed, CP2C19*2 and -*3 and VKORC1 G-1639A polymorphisms are strongly associated to warfarin sensitivity and bleeding risk. Such genetic variants are even more important in presence of drug drug interactions.
Therefore, the pharmacogenetic issue should be introduced as a study limitation or at least in the discussion.
The Authors may introduce references in this field:
Mazzaccara et al. Warfarin anticoagulant therapy: a Southern Italy pharmacogenetics-based dosing model. PLoS One. 2013 Aug 26;8(8):e71505.
Johnson et al. Clinical Pharmacogenetics Implementation Consortium Guidelines for CYP2C9 and VKORC1 genotypes and warfarin dosing. Clin Pharmacol Ther. 2011 Oct;
Response 1: Thank you for your comments: Role of genetic influence leading to study limitation has been added at the end of discussion in revised manuscript (lines (271-276) along with suggested references (27-28).
Reviewer 2 Report
Major problems:
1- Methods: line 50: the reference to The American College of Cardiology and the American Heart Association (ACC/AHA) guidelines should be provided.
2-Methods: the indication and exact timing of antibiotic administration after prostethic valve replacement should be indicated along with mean warfarin dosage and frequency of INR measurement (eg daily? weekly?) should be indicated.
3- line 90: the number of patients excluded for missing INR values should be indicated.
4- methods. the classification of major and minor bleeding should be indicated and whether such events were classified by independent investigators unaware of the INR values.
4-lines 129-130: reversal of supratherapeutic INR values is usually performed with vitamin K in the absence of bleeding or in case of minor but also major bleeding, while FFP is usually indicated in case of supratherapeutic INR values with major bleeding.
5- Results: the type and frequency of bleeding events, both minor and major should be reported in each of the three groups of patients.
6- References: ref n15: it should be updated as it refers to guidelines issued in 2008. The most recent ones were published in 2017 with indication of Anticoagulation with a VKA (vitamin K antagonists) to achieve an INR of 2.5 recommended for patients with a mechanical bileaflet or current-generation single-tilting disc AVR and no risk factors for thromboembolism I B
Anticoagulation with a VKA is indicated to achieve an INR of 3.0 in patients with a mechanical AVR and additional risk factors for thromboembolic events (AF, previous thromboembolism, LV dysfunction, or hypercoagulable conditions) or an older-generation mechanical AVR (such as ball-in-cage).178
2014 recommendation remains current. I B
Anticoagulation with a VKA is indicated to achieve an INR of 3.0 in patients with a mechanical MVR.I B. The authors should discuss these indications as they evaluated patients undergoing prosthetic valve surgery in 2017-2019.
Author Response
Thank you for your comments .
Point 1: - Methods: line 50: the reference to The American College of Cardiology and the American Heart Association (ACC/AHA) guidelines should be provided.
Response 1. Thank you for your comments: Reference has been added. (reference number 7)
Point 2-Methods: the indication and exact timing of antibiotic administration after prostethic valve replacement should be indicated along with mean warfarin dosage and frequency of INR measurement (eg daily? weekly?) should be indicated.
Response 2. Indications are given as (Lines 101-103): Post-operative discharged patients of both genders (20-65 years) with Mitral valve replacement (MVR), Double valve replacement (DVR) and Aortic valve replacement (AVR) due to various valvular diseases discharged on warfarin therapy were chosen for study.
Line 114 added. All 75 patients discharged with warfarin received standard 5 mg dosages.
Please see Lines 117-119 for frequency of INR measurement. A documented therapeutic INR before the start of quinolones and cefixime at the time of discharge and a documented INR within 7 days (at 6 or 7 day of being discharged) after it has been started was recorded
Point 3- line 90: the number of patients excluded for missing INR values should be indicated.
Response 3: Eight patients were excluded from the study, added in method (line 110-111).
Point 4- methods. the classification of major and minor bleeding should be indicated and whether such events were classified by independent investigators unaware of the INR values.
Response 4. There were few patients who experienced major bleeding events and all the complications were documented by consultant cardiac surgeon who were unaware of the study. Therefore, their bleeding data was not taken.
Point 4-lines 129-130: reversal of supra therapeutic INR values is usually performed with vitamin K in the absence of bleeding or in case of minor but also major bleeding, while FFP is usually indicated in case of supra therapeutic INR values with major bleeding.
Amendment has been done as suggested and it can be seen in revised manuscript (lines 161-165).
5- Results: the type and frequency of bleeding events, both minor and major should be reported in each of the three groups of patients.
Response 5. We did not collect the data of bleeding events unfortunately, since complications were handled by cardiac surgeons unaware of the study. However, we collected the data of INR changes which were significant
6- References: ref n15: it should be updated as it refers to guidelines issued in 2008. The most recent ones were published in 2017 with indication of Anticoagulation with a VKA (vitamin K antagonists) to achieve an INR of 2.5 recommended for patients with a mechanical bileaflet or current-generation single-tilting disc AVR and no risk factors for thromboembolism I B
Anticoagulation with a VKA is indicated to achieve an INR of 3.0 in patients with a mechanical AVR and additional risk factors for thromboembolic events (AF, previous thromboembolism, LV dysfunction, or hypercoagulable conditions) or an older-generation mechanical AVR (such as ball-in-cage).178
2014 recommendation remains current. I B
Anticoagulation with a VKA is indicated to achieve an INR of 3.0 in patients with a mechanical MVR.I B. The authors should discuss these indications as they evaluated patients undergoing prosthetic valve surgery in 2017-2019.
Response 6. New guidelines as suggested by reviewer has been added (lines 192-200) in revised manuscript along with updated 2017 reference. Reference number 17.
Reviewer 3 Report
This manuscript describes potential interactions between quinolone antibiotics and warfarin in patients that underwent valve surgery. The authors need to display their demographic data by the patient groups receiving the different antibiotics. It is not possible with the methods currently used to know the number of patients per group, if the three groups have similar age, type of surgery and comedications. Thus the validity of the author's conclusions is unreliable unless the data is presented in a more transparent manner.
Time of enrolment in each group post surgery would also be interesting, were the patients stable on warfarin at enrolment or had dosing just been initiated. It is known that often the first weeks of warfarin dosing can result in large INR variations.
Authors should also more clearly describe what is known in the literature about possible interactions with warfarin and quinolones, and compare this to their findings.
Author Response
Thank you for your comments. The demographic data of three groups has been elaborated in results in revised manuscript as shown in lines 127-134. While, moxifloxacin group of patients were having ages between 24-63y, cefixime group patients between 29-64y and levofloxacin group between 28-72y.
Total 12 patients with aortic stenosis(AS), 3 with mitral regurgitation (MR), 2 aortic stenosis and aortic regurgitation (AS/AR), 2 aortic stenosis and mitral stenosis(AS/MS), 4 MS, 1 AR and 1 MS/MR were presented in moxifloxacin group.
In cefixime group 11 patients were AS, 1 with AS/MS, 1 with MR, 8 MS, 3 AR and 1 MR/MS.
Levofloxacin group patients were 11 with AS, 10 MS,1 AS/MR, 1 AS/MR/AR,1 MR and 1 AS/MS.
Discharged patients were selected when they were stabilized already with their INR in therapeutic range with warfarin after surgery in wards. They were then discharged with 5mg warfarin dose with two quinolones or cefixime along with concomitant medications showing no interaction with warfarin in literature.
Literature findings about possible interaction are mentioned in discussion (Lines 259-265) as: It has been reported that ciprofloxacin and ofloxacin shows interaction with warfarin, leading to elongation in PT. Although few case studies including two new cases of warfarin-moxifloxacin interactions were reported and added to the previously published 12 case reports and therefore highlight the importance of such interaction. No statistically remarkable variations in the INR values observed before and those after levofloxacin administration in one study were found in correlation with other studies. However, in 2009 the retrospective assessment of potential interaction between levofloxacin and warfarin was conducted and a correlation was found between elevation of INR values because of levofloxacin addition to warfarin therapy but does not clarify the severity of interaction. This was further demonstrated by few number of patients studied and changes in INR values resulting from various variables such as type of diet taken, concurrent drug therapy, abnormal liver function tests, low albumin levels, fever, illness and stress that could affect warfarin pharmacokinetics or pharmacodynamics’